# Verifiable measurement-based quantum random sampling with trapped ions

Martin Ringbauer [1] ✉, Marcel Hinsche [2], Thomas Feldker[1,3],
Paul K. Faehrmann[2], Juani Bermejo-Vega[2,4,5], Claire L. Edmunds [1],
Lukas Postler[1], Roman Stricker [1], Christian D. Marciniak [1], Michael Meth[1],
Ivan Pogorelov [1], Rainer Blatt[1,3,6], Philipp Schindler [1], Jens Eisert [2,7,8],
Thomas Monz [1,3] & Dominik Hangleiter [9,10] ✉

Quantum computers are now on the brink of outperforming their classical counterparts. One way to demonstrate the advantage of quantum computation is through quantum random sampling performed on quantum computing devices. However, existing tools for verifying that a quantum device indeed performed the classically intractable sampling task are either impractical or not scalable to the quantum advantage regime. The verification problem thus remains an outstanding challenge. Here, we experimentally demonstrate efficiently verifiable quantum random sampling in the measurement-based model of quantum computation on a trapped-ion quantum processor. We create and sample from random cluster states, which are at the heart of measurement-based computing, up to a size of 4 × 4 qubits. By exploiting the structure of these states, we are able to recycle qubits during the computation to sample from entangled cluster states that are larger than the qubit register. We then efficiently estimate the fidelity to verify the prepared states—in single instances and on average—and compare our results to cross-entropy benchmarking. Finally, we study the effect of experimental noise on the certificates. Our results and techniques provide a feasible path toward a verified demonstration of a quantum advantage.

In *quantum random sampling*, a quantum device is used to produce samples from the probability distribution generated by a random quantum computation[1]. This is a particularly challenging task for a classical computer asymptotically[2–4] and in practice[5,6] and thus at the center of recent demonstrations of a quantum advantage[7–10]. A key challenge for such experiments, however, is to verify that the produced samples indeed originate from the probability distribution generated by the correct random quantum computation. Verification based only on classical samples from the device is fundamentally inefficient[11]. In practice, the verification problem has been approached using so-called linear *cross-entropy benchmarking* (XEB)[7,12]. The corresponding XEB *score* is obtained by averaging the ideal probabilities corresponding to the observed experimental samples. XEB is appealing since it has been argued that even achieving any non-trivial XEB

[1]Universität Innsbruck, Institut für Experimentalphysik, Innsbruck, Austria. [2]Dahlem Center for Complex Quantum Systems, Freie Universität Berlin, Berlin, Germany. [3]Alpine Quantum Technologies GmbH, Innsbruck, Austria. [4]Departamento de Electromagnetismo y Física de la Materia, Avenida de la Fuente Nueva, Universidad de Granada, Granada, Spain. [5]Institute Carlos I for Theoretical and Computational Physics, Campus Universitario Fuentenueva, Granada, Spain. [6]Institut für Quantenoptik und Quanteninformation, Österreichische Akademie der Wissenschaften, Innsbruck, Austria. [7]Helmholtz-Zentrum Berlin für Materialien und Energie, Berlin, Germany. [8]Fraunhofer Heinrich Hertz Institute, Berlin, Germany. [9]Joint Center for Quantum Information and Computer Science (QuICS), University of Maryland & NIST, College Park, MD, USA. [10]Joint Quantum Institute (JQI), University of Maryland & NIST, College Park, MD, USA. ✉e-mail: martin.ringbauer@uibk.ac.at; mail@dhangleiter.eu

score might be a classically computationally intractable task[13,14] and that it can be used to sample-efficiently estimate the *quantum fidelity* of the experimental quantum state[7,15]. However, XEB requires a classical simulation of the implemented circuits to obtain the ideal output distribution. The computational run-time of estimating XEB from samples thus scales exponentially, rendering it practically infeasible in the quantum advantage regime. Moreover, it is not always a good measure of the quantum fidelity[16–18]. Another way classical verification of quantum devices has been approached is via interactive proof systems[19,20], albeit at the cost of large device overheads[21,22]. Hence, classical approaches to verification have limited applicability for devices operating in the quantum advantage regime.

These challenges raise the question of whether there are quantum verification techniques that could be used to efficiently verify quantum random sampling experiments, even when their simulation is beyond the computational capabilities of classical devices. Answering this question in the affirmative, we turn to a different universal model of quantum computation—measurement-based quantum computing (MBQC)[23,24]. In contrast to the circuit model, a computation in MBQC proceeds through measurements, instead of unitary operations, applied sequentially to an entangled *cluster state*[24]. Roughly speaking, a cluster state on an $n \times m$ grid of qubits can be used to execute an $n$-qubit, depth-$m$ quantum circuit. Appropriately randomized, cluster states turn out to be a source of random samples appropriate for demonstrating a quantum advantage via random sampling[25–27]. Crucially, though, each cluster state is fully determined by a small set of so-called *stabilizer operators*. By measuring the stabilizer operators using well-characterized single-qubit measurements, preparations of these cluster states can be efficiently verified[28–33].

Here, we experimentally demonstrate efficiently verifiable quantum random sampling in the MBQC model in two trapped-ion quantum process*ors* (TIQP). While cluster state generation in TIQP has previously been limited to a size of $2 \times 2$[34], we overcome this limitation with a two-fold approach. First, we use pairwise addressed Mølmer-Sørensen entangling operations[35,36] in a fully connected linear chain to enable the efficient generation of clusters up to a size of $4 \times 4$ qubits.

Second, we make use of spectroscopic decoupling and optical pumping[37] to perform mid-circuit readout and reset of qubits in order to recycle them. In this way, we are able to sequentially measure rows of the cluster and then reuse the measured qubits to prepare a new row of the cluster, while maintaining entanglement with the remaining qubits, see Fig. 1c. This allows us to sample from a cluster state on a lattice that is larger than the size of the physical qubit register. This combination of techniques provides a feasible path towards generating large-scale entangled cluster states using trapped ions.

We then estimate the fidelity of the experimental cluster states in order to verify those states. Specifically, we apply a novel variant of direct fidelity estimation (DFE)[28,32] to estimate the single-instance fidelity of a fixed cluster state, and the average fidelity of random cluster states. The single-instance fidelity certifies the samples from a fixed, random cluster state, and therefore a quantum advantage for sufficiently large cluster states[29]. Conversely, the average fidelity of random cluster states is a benchmark of the average performance of the quantum processor in the quantum advantage regime[38]. Direct (average) fidelity estimation is therefore a unified framework for verification and benchmarking of MBQC, analogously to XEB. However, in contrast to XEB, the fidelity estimation approach has several major advantages: First, it is efficient in terms of both the required number of experiments, and the complexity of the postprocessing. Second, it requires knowledge only of the measurement noise as opposed to the noise properties of all gates which is required for XEB[16–18]. Finally, the fidelity gives a rigorous bound on the quality of the samples from a single quantum state, whereas XEB is generally only accurate on average.

In order to assess the performance of the fidelity-derived certificates, we compare them to the available—but inefficient—classical means of certification of the samples, which is still possible in our proof-of-principle demonstration. In the single-instance case, we compare the experimental performance of the single-instance fidelity estimate to the empirical total-variation distance (TVD) of the sampled distribution. In the average case, we compare the average fidelity

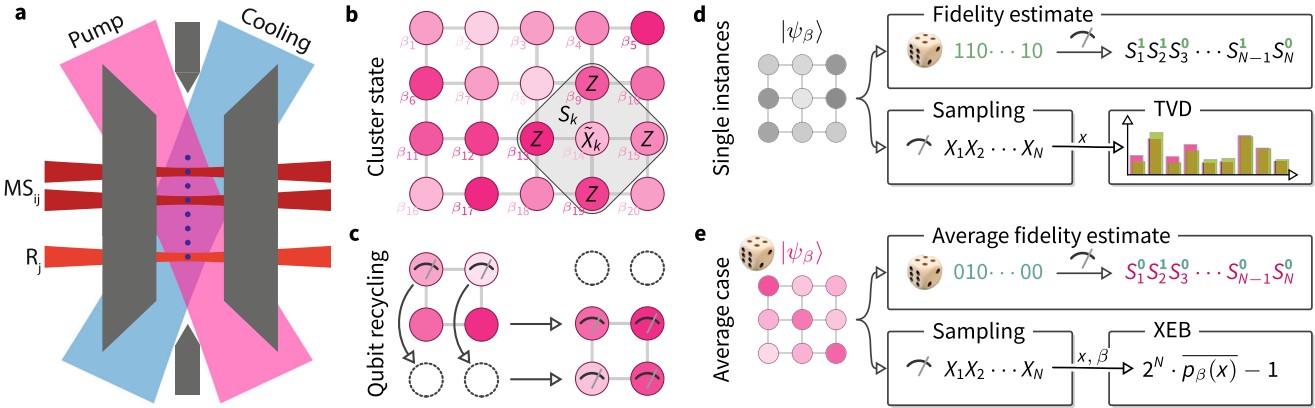

**Fig. 1 | Overview of the experiment. a** Sketch of the ion trap quantum processor. Strings of up to 16 ions are trapped in a linear chain. Any single ion or pair of ions can be individually addressed by means of steerable, tightly focused laser beams (dark red) to apply resonant operations $R_j$ or Mølmer-Sørensen entangling gates $MS_{i,j}$. Global detection, cooling (blue), and repumping (pink) beams are used to perform a mid-circuit reset of part of the qubit register[35]. **b** Implemented cluster states. Cluster states with local rotation angles $\beta_i \in \{0, \frac{\pi}{4}, \ldots, \frac{7\pi}{4}\}$ up to a size of $4 \times 4$ qubits are created in the qubit register. Each cluster state is defined by its $N$ stabilizers $S_k$ which are given by rotated $X$ operators $\bar{X}_k = X_k(\beta_k)$ at each site $k = 1, \ldots, N$ multiplied with $Z$ operators on the respective neighboring sites. **c** Recycling of qubits. Using sub-register reset of qubits, we prepare cluster states that are larger than the qubit register. For example, using four ions, we prepare cluster states of size $2 \times 3$. **d** Single-instance verification. In order to verify single

cluster state preparations with fixed rotation angles $\beta$, we measure it in different bases. To perform fidelity estimation we measure uniformly random elements of its stabilizer group, which is obtained by drawing a random product of the $N$ stabilizers $S_k$, indexed by a length-$N$ random bitstring indicating for each $S_k$ whether it participates in the product. To sample from the output distribution, we measure in the $X$-basis. These samples are verified in small instances by the empirical total-variation distance (TVD). **e** Average-case verification. To assess the average quality of the cluster state preparations, we perform measurements on cluster states with random rotations. By measuring a random element of the stabilizer group of each random cluster state, we obtain an estimate of the average fidelity. From the samples from random cluster states in the $X$-basis, we compute the *cross-entropy benchmark* (XEB) by averaging the ideal probabilities $p_\beta(x)$ corresponding to the samples $x$ and the cluster with angles $\beta$.

estimate to the average XEB score. Additionally, we study the effect of native noise sources on the different measures of quality.

Our work thus provides a clear and feasible path towards verified quantum advantage. It does so by developing a new approach to verifying random cluster states based on a variant of DFE, introducing the use of qubit recycling in order to generate large clusters, and demonstrating the feasibility of the proposed techniques in the presence of experimental noise.

## Results

### Sampling and verification protocols

In the circuit model, natural examples of random computations are, for instance, circuits composed of Haar-random two-qubit gates[39], or composed of native entangling gates and random single-qubit gates[7]. In contrast, in MBQC, a universal quantum computation can be realized by adaptively performing single-qubit rotations around the $Z$-axis on a cluster state and measuring in the Hadamard basis conditioned on the outcomes of previous Hadamard-basis measurements[24,27,40]. This leads to a natural notion of random MBQC wherein those single-qubit $Z$-rotations are applied with angles chosen randomly from an appropriate discretization of the unit circle[26,27]. Adaptively performing single-qubit rotations then becomes superfluous since they are chosen randomly anyway, and an outcome pattern on the square lattice defines both, an effective quantum circuit given the random rotations, and the outcomes of measuring that circuit. Hence, repeatedly measuring a fixed, random cluster state without adaptive rotations is equivalent to measuring many different quantum circuits chosen randomly from an ensemble defined by the random rotations, see Supplementary Fig. 2.

The largest discretization in the choice of single-qubit rotations leading to a computationally universal MBQC scheme consists of eight evenly spaced angles, corresponding to powers of the $T$ gate. In exactly the same way as for circuit-based sampling schemes[3,12], there is strong complexity-theoretic evidence that for $m \gtrsim n$ approximately reproducing the outcome statistics of such random measurements is classically intractable[26,27]. In fact, in both cases, even producing samples from a quantum state with a non-vanishing or only slowly vanishing fidelity is likely classically hard[16,41]. Quantum advantage aside, the effective computations implemented by random cluster states generate a unitary 2-design[27] and therefore yield a reliable average-performance benchmark for measurement-based computations[42].

Concretely, the MBQC random sampling protocol we apply is then the following[26,27] (see Fig. 1, and Supplementary Note 3 for explicit circuits):

- Prepare a cluster state on $N = n \times m$ qubits on a rectangular lattice by preparing each qubit in the $|+\rangle$ state and applying controlled-$Z$ gates between all neighbors.
- Apply single-qubit rotations $Z(\beta) = e^{-i\beta Z/2}$ with random angles $\beta \in \{0, \frac{\pi}{4}, \ldots, \frac{7\pi}{4}\}$ to every qubit.
- Measure all qubits in the Hadamard basis.

We note that the state preparation steps 1 and 2 can also be achieved by time-evolving an initial state $|+\rangle^{\otimes N}$ under an Ising Hamiltonian on an $n \times m$ lattice with random local fields depending on the $\beta_k$[26,27], but the gate-based approach outlined here is more suitable for TIQP.

Using a variant of DFE, we assess the quality of both single cluster states with local $Z$ rotations, and the average quality of such state preparations. In DFE, we estimate the fidelity $F(\rho, |\psi\rangle\langle\psi|) = \langle\psi|\rho|\psi\rangle$ of a fixed experimental state $\rho$ by measuring random operators from the *stabilizer group* of the random cluster state $|\psi\rangle$ and averaging the results, see Methods for detail. The stabilizer group is the group generated by the $N$ stabilizers of the random cluster. Each stabilizer $S_k$ is the product of a rotated $X$-operator at site $k$—given by $X_k(\beta) = e^{-i\beta X_k/2}$—and $Z$-operators on the neighboring sites, giving rise to a characteristic

star shape on the square lattice, see Fig. 1b. Importantly, all elements of the stabilizer group are products of single-qubit operators. Our trust in the fidelity estimate therefore only depends on our ability to reliably perform single-qubit measurements, which we verify. In order to measure the average fidelity over the set of cluster states, we prepare random cluster states and for each state measure a random element of its stabilizer group. We then average the results to obtain an estimate of the average fidelity. At a high level, fidelity estimation thus exploits our ability to measure the experimental state in different bases. It requires a number of experimental state preparations that is *independent* of the size of the system, making it scalable to arbitrary system sizes, see Methods for details. We note that we also measured a witness for the fidelity[29] and find that it is not practical in a scalable way for noisy state preparations, as we detail in Supplementary Note 1.

Given the relatively small system sizes of the experiments in this work, we are also able to directly compute non-scalable measures of quality that make use of the classical samples only. This enables us to compare fidelity estimation with inefficient classical verification methods in different scenarios. To classically assess the quality of samples from a fixed experimental state preparation, we use the TVD $d_{TV}(P, Q) = \sum_x |P(x) - Q(x)|/2$. The TVD quantifies the optimal probability of distinguishing the experimentally sampled distribution $Q$ and the ideal one for a noiseless cluster $P$. The TVD is the classical analog of the trace distance $d_{Tr}(\rho, |\psi\rangle\langle\psi|) = Tr(|\rho - |\psi\rangle\langle\psi||)/2$, which quantifies the optimal probability of distinguishing the sampled quantum states $\rho$ and $|\psi\rangle\langle\psi|$. The fidelity $F$ upper-bounds the trace distance[43] and therefore the TVD of the sampled distributions as

$$d_{TV} \leq d_{Tr} \leq \sqrt{1-F}. \tag{1}$$

The root infidelity $\sqrt{1-F}$ can therefore be used to certify the classical samples from $\rho$. We note that it is a priori not clear how tight this bound is in an experimental scenario and how experimental noise affects the different verification methods. In order to classically assess the average quality of the quantum device, we estimate the linear XEB fidelity between $Q$ and $P$, which is defined as $f_{lin}(Q, P) = 2^n \sum_x Q(x)P(x) - 1$[1]. The average XEB fidelity over the random cluster states measures the average quantum fidelity in the regime of low noise[16-18], see Supplementary Note 5.

### Experimental implementation

We implement the random MBQC sampling and verification protocols in two ion-trap quantum processors. Quantum information is encoded in the $S_{1/2}$ ground state and $D_{5/2}$ excited state of up to 16 $^{40}Ca^+$ ions confined in a linear Paul trap[36,37]. We use these devices to implement two sets of experiments. First, we generate rectangular $n \times m$ random cluster states of up to 16 ions by appropriately entangling the respective ions in a linear chain using pairwise addressed Mølmer-Sørensen-gates[35,36]. In a second, proof-of-principle set of experiments on a device with an extended control toolbox yet somewhat lower fidelities, we make use of spectroscopic decoupling and optical pumping to recycle qubits to demonstrate a more qubit-efficient way to sample from large-scale entangled cluster states. By construction, the 2D cluster states require entangling gates between neighboring qubits only. As a consequence, when generating the cluster from top to bottom, once the first row has been entangled to the second, we can measure the qubits of the first row. Once measured, these qubits can be reset to the ground state, prepared in their appropriate initial states and entangled as the third row of the cluster state, and so on. Due to the local entanglement structure of the cluster state, the measurement statistics obtained in this way are identical to the statistics that would be obtained from preparing and measuring the full cluster state at once.

Experimentally, we make use of mid-circuit readout capabilities[44] using an EMCCD camera to read out a subset of the qubits, while

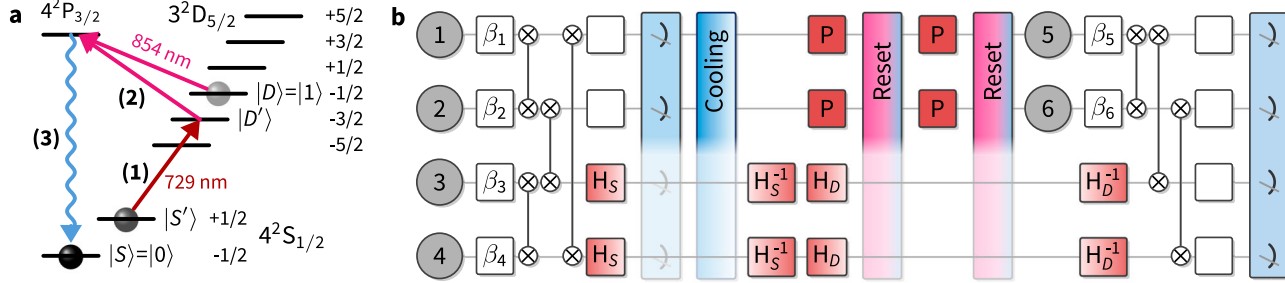

**Fig. 2 | Sketch of the circuit with qubit recycling. a** Recycling. After detection, a measured qubit is either still in the $|1\rangle$ state ("dark" outcome) or in one of the two S levels ("bright" outcome). We reset it to the $|0\rangle$ state by first applying an addressed $\pi$-pulse (1) on the $|S'\rangle \rightarrow |D'\rangle$-transition. A subsequent global 854 nm quench pulse (2) transfers population from all D-levels to the $P_{3/2}$ manifold, from which (3) spontaneous decay occurs, preferentially to the $|0\rangle$ state in the S manifold. We repeat this process twice, which is sufficient to return about 99% of the population to the $|0\rangle$ state. **b** Circuit. The individual qubits are prepared in a product state depending on the random angles $\beta_i$ and entangled via *XX* interactions and some single-qubit gates (white boxes) to create a cluster state; see Supplementary Note 3. The measurement of the qubits is achieved by exciting the $P \leftrightarrow S$ transition. In order to perform a circuit with recycling, a coherent $\pi$-pulse on the $S \rightarrow D'$ transition

(denoted by $H_S$) is applied to `hide' the qubits which should not be measured in the D-manifold. After the measurement, the chain is cooled using polarization-gradient cooling. The reset makes use of local pulses on the measured qubit that transfer the remaining population of $|S'\rangle$ to the $D_{5/2}$-manifold (denoted by P) and global pulses that transfer the population of that manifold back to $|0\rangle$. Prior to the reset, all unmeasured qubits are `hidden' in the $S_{1/2}$-manifold. For this, the population which was in $|0\rangle$ prior to the measurement is coherently transferred back to $|S\rangle$ via a $\pi$-pulse ($H_S^{-1}$), and the population which is in $|1\rangle$ is transferred to $|S'\rangle$ via a $\pi$-pulse on the $D \rightarrow S'$ transition ($H_D$). After the reset procedure (**a**), a $\pi$-pulse ($H_D^{-1}$) is applied to the unmeasured qubits to transfer the population which was previously in $|1\rangle$ back from $S'$.

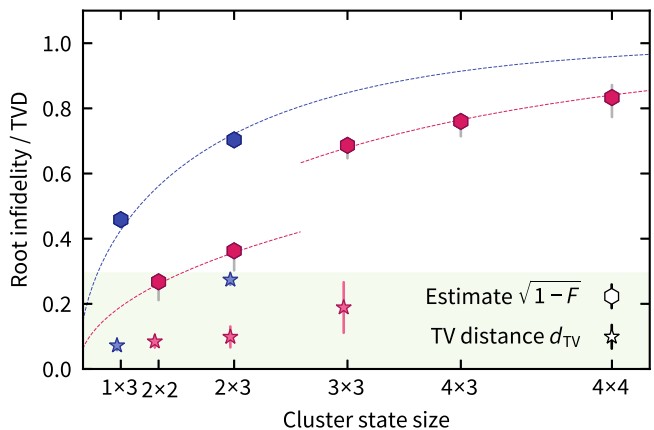

**Fig. 3 | Experimental results for single-instance verification.** Root infidelity estimate $\sqrt{1-F}$ (hexagons), and empirical TVD (stars) for single instances of random MBQC cluster states with recycling (blue) and without (pink). Note that the horizontal axis is labeled with the cluster size $n \times m$ and scaled with qubit number $nm$. The root infidelity upper-bounds the TVD per Eq. (1). Colored error bars represent the $3\sigma$ interval of the statistical error. Uncorrelated measurement noise reduces or increases the measured state fidelity compared to the true fidelity asymmetrically depending on its value, such that the shown values are lower bounds to the true state fidelity, see the Methods section for details. The worst-case behavior of the measurement noise is represented by gray error bars. In the non-recycling experiment, the register size is increased between the $2 \times 3$ and the $3 \times 3$ instance, leading to a decrease in the local gate fidelities. Modeling the noise as local depolarizing noise after each entangling gate (dotted lines), we obtain effective local Pauli error probabilities after the two-qubit gates of 5.3%, 2.6%, and 1.0%, for the recycling data, for the large-register non-recycling data, and the small register non-recycling data, respectively; see Supplementary Note 5. The shaded green area is the acceptance region corresponding to an infidelity threshold of 8.6% arising in the rigorous hardness argument as sketched in Supplementary Note 4. Since the accuracy of the TVD estimate scales with the system dimension already for cluster sizes of $4 \times 3$ and $4 \times 4$ infeasible amount of samples would be required for an accurate estimate, and hence these are not shown. See Supplementary Table 1 for experimental details.

spectroscopically decoupling the remaining qubits from the readout beams, see Fig. 2. After the readout, we re-cool the ion string using a combination of Doppler cooling and polarization-gradient cooling for a total of 3 ms. Then we employ two rounds of optical pumping using addressed 729 nm pulses in combination with a global 854 nm quench

beam to reset the qubits to the $|0\rangle$ ground state[37], while the remaining qubits are spectroscopically decoupled. This completes the reset and we can now prepare the measured qubits in their new states and entangle them to the remaining qubits of the cluster, see Fig. 2. This procedure enables us to sample from entangled quantum states with more qubits than the physical register size of the used quantum processor. Specifically, to prepare an $n \times m$ cluster state at least $n + 1$ repeatedly recycled qubits are required, and the required circuit depth (and recycling steps) decreases as the number of available physical qubits increases.

For every state, we perform sampling and verification measurements. We measure the state in the Hadamard basis in order to perform sampling. For verification, we measure a random element of its stabilizer group. When verifying a single instance of a state preparation, we repeat this procedure for a fixed state and then estimate the fidelity from the random stabilizer measurements and the TVD from the classical samples. To estimate the average performance of the device, we repeat the procedure for random states and estimate the average fidelity and the average XEB fidelity, see Fig. 1d, e. Finally, for the $2 \times 2$ cluster, we study the effect of increasing global (local) dephasing noise on the verification performance by adding small (un)correlated random $Z$-rotations before and after each entangling gate.

## Experimental results

We first measure the fidelity and TVD of single random cluster state preparations for various cluster sizes. The results demonstrate that the root-infidelity provides meaningful upper bounds on the TVD, see Fig. 3. Importantly, while the efficiently measurable and computable root infidelity estimate is guaranteed to bound the TVD per Eq. (1), these scalable bounds are not tight. This is seen in Fig. 3 as a gap between the root infidelity upper bound and the measured TVD values. Indeed, it is expected that reproducing the full quantum state (as measured by the fidelity) is a more stringent requirement than merely reproducing the outcome distribution in one particular measurement basis (as measured by the TVD). Hence, the efficient quantum methods require higher fidelities for the corresponding certificate to meet the quantum advantage threshold. Notably, above the cluster size of $3 \times 3$ qubits, empirically estimating the TVD with sufficient accuracy is practically infeasible due to the exponentially growing state space. In the proof-of-principle experiments, where recycling is used, we see the same qualitative behavior, although the overall root infidelities are

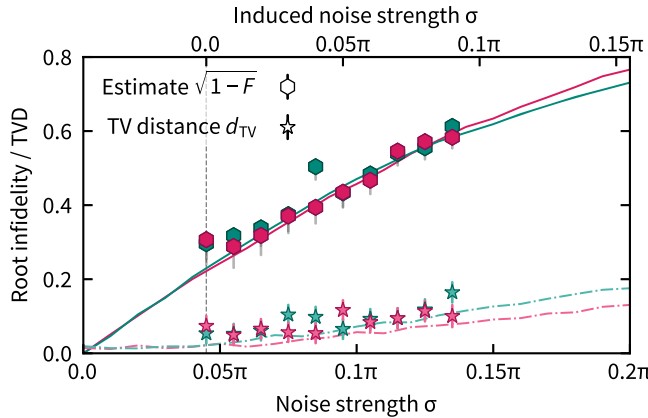

**Fig. 4 | Single-instance verification with artificially added phase noise.** Root infidelity estimate $\sqrt{1-F}$ (hexagons) and empirical total-variation distance $d_{TV}$ (stars) of a $2 \times 2$ cluster state with artificially introduced local (pink) and global (green) phase noise—$Z$-rotations with rotation angle drawn from a Gaussian distribution with variance $\sigma^2$—before and after Mølmer-Sørensen gate applications as a function of the noise strength $\sigma$, see Methods for details. Solid (dashed) lines show simulated root infidelity (total-variation distance) for the respective types of noise. The experimental data (top axis) is shifted with respect to the simulations (bottom axis) due to the fact that there is residual noise when no artificial noise is introduced. The value of the relative shift given by $0.045\pi$ (dashed vertical line) provides an estimate for the natural noise strength. Colored error bars represent the $3\sigma$ interval of the statistical error. The systematic measurement error of the fidelity estimate is represented by gray error bars.

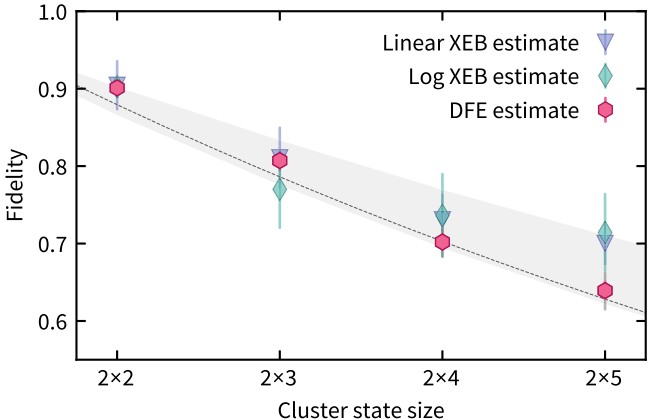

**Fig. 5 | Experimental results for average performance verification.** Average fidelity estimate from direct fidelity estimation (DFE) (pink hexagons), from linear XEB (triangles), and from logarithmic (log) XEB (diamonds, see Methods for the definition) using 1000 random cluster states and 50 shots per state. Based on calibration data for the gate fidelities of single-qubit gates $f_{1Q} = 99.8\%$, two-qubit gates $f_{2Q} = 97.5 \pm 0.5\%$, and measurements $f_M = 99.85\%$, we compute a prediction for the fidelity (gray shaded line). We extract an effective local Pauli error probability of 1.7% (dotted line), see Supplementary Note 5. Colored error bars represent the statistical $3\sigma$ error. For uncorrelated measurement noise, the fidelity estimate provides a lower bound to the true state fidelity. Gray error bars represent the worst-case systematic measurement error.

higher. This is likely due to imperfect re-cooling, which only cools the system to low motional occupation of $\bar{n} \sim 2$ phonons. While the *Mølmer-Sørensen* (MS) gate is insensitive to the motional occupation to first order[45], higher phonon number leads to a larger sensitivity to calibration errors. Moreover, the recooling process takes 3 ms, during which the system experiences some dephasing. Hence, the recycling and non-recycling experiments are not directly comparable. It is,

however, anticipated that the technical limitations can be overcome through the use of mid-circuit ground-state cooling and faster recycling schemes, such that comparable fidelities between the two methods can be achieved, as would also be required for realizing quantum error correction.

Figure 4 shows the results of the fidelity and TVD measurements for an increasing amount of noise on the $2 \times 2$ cluster state in comparison to numerical simulations. We observe an increasing gap between TVD and upper bound from the root infidelity estimate (cf. Eq. (1)) with the amount of noise in a fixed quantum circuit. These results indicate that output distributions of states subject to a significant amount of dephasing noise may still have a TVD well below the root infidelity. Comparing the experimental results with the simulations also allows us to deduce the natural noise floor in the experiment.

We then measure the fidelity of cluster state preparations, averaged over the random circuits, and show the results in Fig. 5. We compare the fidelity estimates to the classical estimates of fidelity via XEB depending on the relative dimensions of the cluster, since in the circuit model the quality of XEB as a fidelity estimator depends on the circuit depth[17,18]. We observe a consistently larger variance of the XEB estimate of the fidelity than of the direct fidelity estimate, and deviates for the $2 \times 5$ cluster. This may be due to the fact that the XEB fidelity depends on the type and strength of the experimental noise, but also the specific dimensions of the cluster state and the effective circuit ensemble[16–18]. Hence, while XEB generally seems to reflect the order of magnitude of the true fidelity, extreme care must be taken when using the XEB as an estimator of the fidelity.

## Discussion

We conclude that direct (average) fidelity estimation provides an efficient and scalable means of certifying both single instances and the average quality of measurement-based computations. This is the case since the sample complexity of the fidelity estimate for arbitrary generalized stabilizer states is independent of the size of the system and the postprocessing is efficient. Larger systems can therefore be verified with the same number of experiments as we have performed.

More generally, our results demonstrate that the measurement-based model of quantum computation provides a viable path toward efficient verification of quantum random sampling experiments, which is not known to be possible in the circuit model. In particular, all known methods for fidelity estimation[28,46] in general scale exponentially with the number of qubits. We also note that, although MBQC is formally equivalent to the circuit model, relating a quantum circuit to an MBQC requires a space-time mapping and a feedforward procedure. Hence, our verification protocol at the level of the cluster state has no direct analog in circuit-based computations. While the experiments in this work are still far from the quantum advantage regime, we have successfully demonstrated how to use qubit recycling to perform large-scale MBQC with a qubit number that can be quadratically larger than the used ion register. This will enable TIQP comprising on the order of 100 ions and depth 50 to achieve a fully verified quantum advantage in sampling from cluster states with more than $50 \times 50$ nodes.

Besides trapped ions, several other platforms are compelling candidates for demonstrating a verifiable quantum advantage via random cluster state sampling. Examples include arrays of Rydberg atoms in optical tweezers, where the creation of large atom arrays[47] has recently been demonstrated. Another leading platform for cluster state generation is photonics[48], and continuous-variable optical systems where cluster states with up to 30,000 nodes have been experimentally prepared[49,50]. Currently, these states are still Gaussian states and therefore not useful for quantum computing, but it is intriguing to think about how the non-standard topologies of continuous-variable cluster states might be exploited. Traditionally, such continuous variable systems have been used for boson sampling, rather than quantum circuit sampling. While boson sampling is not a

universal model for computation, its efficient verification is possible for both photon-number[51] and Gaussian[52] input states. In practice, however, the verification measurements are entirely different in type compared to the sampling experiments, requiring a different apparatus. In contrast, for verifying MBQC states as performed in this work, the difference between sampling and verification is only local basis rotations. This makes MBQC a particularly compelling candidate for verifiable quantum random sampling.

## Methods

### Verification protocols

MBQC with cluster states is amenable to various types of verification. In particular, we can perform single-instance verification, that is, verification of a single quantum state using many copies of that state. We also perform average verification, that is, an assessment of the quality of state preparations averaged over the ensemble of measurement-based computations defined by the random choices of single-qubit rotation angles $\beta$. We distinguish classical means of verification in which we only make use of classical samples from the cluster state measured in a fixed (the Hadamard) basis, and quantum means of verification in which we measure the cluster state in various different bases.

**Single-instance verification.** In order to perform single-instance verification we apply DFE[28], which uses single-qubit measurements on preparations of the target state $|\psi\rangle$. Since the target state vector $|\psi\rangle$ for our random sampling problem is a locally rotated stabilizer state, with stabilizer operators $S_i$, $\rho = |\psi\rangle\langle\psi|$ is the projector onto the joint $+1$-eigenspace of its $N$ stabilizers. We can therefore expand $\rho$ as the uniform superposition over the elements of its stabilizer group $\mathcal{S} = \langle S_1, \ldots, S_N \rangle$, where $\langle S_1, \ldots, S_N \rangle$ denotes the multiplicative group generated by $S_1, \ldots, S_N$. The fidelity can then be expressed as

$$F = \frac{1}{2^N} \sum_{s \in \mathcal{S}} \langle s \rangle_\rho = \frac{1}{2^N} \sum_{s \in \mathcal{S}} \sum_{\sigma = \pm 1} \langle \pi_s^\sigma \rangle_\rho \cdot \sigma, \tag{2}$$

where $s = \pi_s^+ - \pi_s^-$ is the eigendecomposition of the stabilizer $s$ into its $\pm 1$ subspaces, and $\langle \cdot \rangle_\rho = \mathrm{Tr}[\rho \cdot]$ denotes the expectation value. This suggests a simple verification protocol where in each run a uniformly random element of $\mathcal{S}$ is measured on $\rho$. Averaging over the measurement outcomes $\sigma$ then gives an unbiased estimate of the fidelity according to Eq. (2). Since the measurement outcomes $\sigma$ are bounded by 1 in absolute value, we can estimate the average up to error $\epsilon$ using a number $M$ of measurements from $\mathcal{S}$ that scales as $1/\epsilon^2$ and is independent of the number of qubits.

We also directly estimate the TVD between the empirical distribution and the ideal distribution. Note that estimating the TVD is sample-inefficient since the empirical probabilities need to be estimated, requiring exponentially many samples[11]. It is also computationally inefficient since the ideal probabilities need to be computed.

**Average-case verification.** We measure the average quality of the cluster state preparations $\rho_\beta$ by their average state fidelity

$$\overline{F} := \mathbb{E}_\beta \left[ \langle \psi_\beta | \rho_\beta | \psi_\beta \rangle \right] \tag{3}$$

with the generalized cluster state $|\psi_\beta\rangle$ with random angles $\beta \in \{0, \frac{\pi}{4}, \ldots, \frac{7\pi}{4}\}^{n \times m}$. Here, $\mathbb{E}_\beta[\cdot]$ denotes the expectation value over random $\beta \in [8]^{nm}$, where we let $[8] = \{1, 2, \ldots, 8\}$ and $[k]^l = [k] \times \cdots \times [k]$ $l$ times.

In order to classically estimate the average state fidelity, one can make use of *cross-entropy benchmarking* (XEB) as proposed by refs. 7,12. XEB makes use of the classical samples from a distribution $Q$ and aims to measure how distinct $Q$ is from a target distribution $P$. The

linear and logarithmic XEB fidelities between $Q$ and $P$ are defined as

$$f_{\mathrm{lin}}(Q, P) := 2^n \sum_x Q(x) P(x) - 1, \tag{4}$$

$$f_{\mathrm{log}}(Q, P) := -\sum_x Q(x) \log P(x), \tag{5}$$

respectively. Letting $P_\beta$ be the output distribution of $|\psi_\beta\rangle$ and $Q_\beta$ the output distribution of $\rho_\beta$ after Hadamard-basis measurements, we can estimate the average state fidelity from the average linear (logarithmic) XEB fidelity

$$\overline{f}_{\mathrm{lin(log)}} := \mathbb{E}_\beta \left[ f_{\mathrm{lin(log)}}(Q_\beta, P_\beta) \right], \tag{6}$$

assuming that the total noise affecting the experimental state preparation $\rho_\beta$ is not correlated with $|\psi_\beta\rangle\langle\psi_\beta|$. In order to estimate the (average) XEB fidelities, we need to compute the ideal output probabilities $P_\beta(x)$ and average those over the observed samples $x$. This renders the XEB fidelities a computationally inefficient estimator of the fidelity. They are *sample-efficient* estimators[32], however, provided that the target distribution $P_\beta$ satisfies the expected exponential shape for deep random quantum circuits (or larger cluster states). That is, to achieve an additive estimation error $\epsilon$, a polynomial number of samples in $n$ and $1/\epsilon$ are required. In Supplementary Note 5, we provide the details of the estimation procedure. To date, XEB is the only available means of practically verifying (on average or in the single-instance) universal random quantum circuits.

Here, we observe that in the measurement-based model of quantum computations *fully efficient* (i.e., computationally and sample-efficiently) average-case verification is possible using single-qubit measurements. In fact, we observe that DFE can be extended to measure the average fidelity of random MBQC state preparations. To this end, we observe that the average state fidelity (3) can be expressed analogously to Eq. (2) as

$$\overline{F} = \frac{1}{2^{nm}} \frac{1}{8^{nm}} \sum_{\beta \in \frac{\pi}{4} \cdot [8]^{nm}} \sum_{s_\beta \in \mathcal{S}_\beta} \sum_{\sigma = \pm 1} \langle \pi_{s_\beta}^\sigma \rangle_{\rho_\beta} \cdot \sigma, \tag{7}$$

where $\mathcal{S}_\beta$ denotes the stabilizer group of the locally rotated cluster state $|\psi_\beta\rangle$ with rotation angles $\beta$, $\pi_{s_\beta}^\sigma$ is the projector onto the $\sigma$-eigenspace of $s_\beta \in \mathcal{S}_\beta$, and. Hence, in order to estimate the average state fidelity with respect to the choice of $\beta$, we simply need to sample uniformly random rotation angles $\beta$, and elements $s_\beta$ from the stabilizer group $\mathcal{S}_\beta$ and then measure $s_\beta$ on the state preparation $\rho_\beta$ of $|\psi_\beta\rangle$, yielding outcome $\sigma \in \{\pm 1\}$. Averaging over those outcomes yields an estimator of the average state fidelity with the same sample complexity as DFE has for a single instance. As discussed below, the only assumption required to trust the validity of the result is that the noise in the local single-qubit measurements does not behave adversarially. DFE and direct average fidelity estimation thus provide a unified method for efficiently assessing the single-instance quality and the average quality of MBQC state preparations.

### Finite sampling and error bars

When performing DFE of a fixed cluster state, the simplest protocol is to sample an element $s \in \mathcal{S}$ of the stabilizer group uniformly at random and measure $s$ once; cf. Eq. (2). In this case, the samples are distributed binomially with ideal probability $p = \sum_s \langle \pi_s^\sigma \rangle_\rho / 2^N$, and the error on the mean estimation is given by the standard deviation of the observed binomial distribution. However, in practice, it is much cheaper to repeat a measurement of a stabilizer than to measure a new stabilizer, which requires a different measurement setting. This is why we estimate the fidelity according to the following protocol. We sample $K$ stabilizers uniformly at random and measure each of them $M$ times,

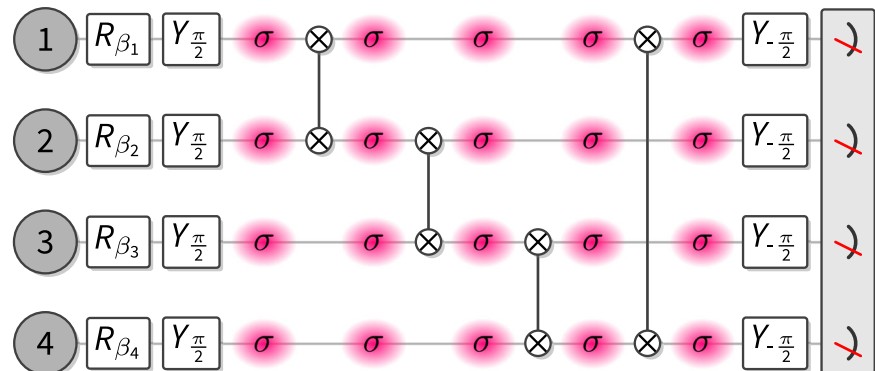

**Fig. 6 | Noisy circuits for the 2 × 2 cluster.** Dephasing noise is simulated by adding random (virtual) $Z$ rotations on all qubits after the initial state preparation and after each MS gate, see Methods. This amounts to roughly equidistant time steps. The rotation angles for the $Z$ rotations are drawn randomly from a normal distribution with zero mean and standard deviation $\sigma \in [0, 0.2\pi]$ every 50 shots. For correlated noise, the parameters in each time step are chosen equally and for uncorrelated noise, they are chosen independently.

obtaining an empirical estimate of the conditional expectation value $\mathbb{E}[\sigma|s] = \sum_{\sigma = \pm 1} \mathrm{Tr}[\rho \pi_s^\sigma] \sigma$. In Supplementary Note 6, we show that the variance of the fidelity estimator $\hat{F} = (KM)^{-1} \sum_{i=1}^{K} \sum_{j=1}^{M} \sigma_{i,j}$, where $\sigma_{i,j}$ is the outcome of measuring stabilizer $s_i$ the $j$th time, is given by

$$\mathrm{Var}[\hat{F}] = \frac{4}{KM}(\mathbb{E}[p_s](1 - \mathbb{E}[p_s])) + \frac{4}{K}\left(1 - \frac{1}{M}\right)\mathrm{Var}[p_s]. \tag{8}$$

Here, the expectation value and variance are taken over $s \in \mathcal{S}$ and $p_s = \mathrm{Tr}[\rho \pi_s^{+1}]$, respectively. Furthermore, the same results carry over to the average fidelity estimate, since sampling from the stabilizer group $\mathcal{S}$ of a single cluster state is now replaced by sampling a random choice of angles $\beta$, and random element of the corresponding stabilizer group $\mathcal{S}_\beta$, not altering the variance.

Eq. (8) gives rise to an optimal choice of $K$ and $M$ for a fixed total number of shots $K \times M$, depending on the expectation value and variance of the stabilizer values $p_s$ and the experimental trade-off between repetitions of the same measurement and changing the measurement setting. In particular, if the distinct elements of the stabilizer group have a small variance over the imperfect state preparation $\rho$, a larger choice of $M$ might be advantageous. In practice, for the instances in which we have abundant data, we subsample the data in order to remain in the situation $M = 1$ of Eq. (2), while in the case of sparse data, we make use of a larger number of shots $M$ per stabilizer.

The variance of the estimate of the XEB fidelity is also given by the law of total variance, generalizing Eq. (8), and spelled out in detail in Supplementary Note 6. Finally, for the TVD, we estimate the error using bootstrapping by resampling given the observed distribution. Specifically, we repeatedly sample from the empirical distribution the same number of times as the experiment and compute the TVD of the samples to the sampled distribution. The resulting TVD follows a Gaussian distribution of which we show the $3\sigma$ interval estimated from 1000 iterations.

**Measurement errors**

A key assumption for the efficient verification of the cluster states prepared here is the availability of accurate, well-characterized single-qubit measurements. A deviation in the measurement directly translates into a deviation in the fidelity estimate, and hence a high measurement error in the worst case translates into a high error in the resulting fidelity estimate. Because the single-qubit measurements we use comprise single-qubit gates followed by readout in a fixed basis, the measurement error has two main contributions: (i) imperfections in the single-qubit rotations for the basis choice, and (ii) imperfections in the readout.

The single-qubit gate errors are well characterized by randomized benchmarking, showing an average single-qubit Clifford error rate of $3 \pm 2 \times 10^{-4}$ [35] for the recycling device and $14 \pm 1 \times 10^{-4}$ [36] for the second device. The native $Z$ measurement is then performed by scattering photons on the short-lived $S_{1/2} \leftrightarrow P_{1/2}$ transition. Ions in the $|0\rangle$ state will scatter photons, while ions in the $|1\rangle$ state remain dark. Hence, there are two competing contributions to the readout error. On the one hand, long measurement times suffer from amplitude damping noise due to spontaneous decay of the $|1\rangle$ state (lifetime ~1.15s) during readout. On the other hand, for short readout times, the Poisson distributions for the two outcomes will start to overlap, leading to discrimination errors. In the experiments presented here, the second contribution is suppressed to well below $10^{-5}$ by using measurement times of 1ms for the recycling device and 2ms on the non-recycling device, leaving only a spontaneous decay error of $<1 \times 10^{-3}$ [37] for the recycling device and $<2 \times 10^{-3}$ for the non-recycling device. Hence, the worst-case readout error is $<1.5 \times 10^{-3}$ per qubit for the recycling device and $<3.5 \times 10^{-3}$ per qubit for the second device. Given the single-qubit readout error $e_1$, the overall measurement error on an $n$-qubit device is then given by $e_M = 1 - (1 - e_1)^n$.

Given a true pre-measurement state fidelity $F$, we consider the effect of the measurement errors on the estimated fidelity $\hat{F}$. In the one extreme case, the measurement errors flip the sign of the stabilizers with value +1 on the pre-measurement state, but keep the sign of those with a −1 outcome, resulting in a reduced state fidelity $\hat{F}_{\min} = 2((1 + F)/2 - e_M \cdot (1 + F)/2) - 1$. In the other extreme case, they flip the sign of only the stabilizers with value −1 on the pre-measurement state yielding $\hat{F}_{\max} = 2((1 + F)/2 + e_M \cdot (1 - F)/2) - 1$. This defines the worst-case error interval for $\hat{F}$ as $[(\hat{F} - e_M)/(1 - e_M), (\hat{F} + e_M)/(1 - e_M)]$.

If on the other hand the measurement errors are benign, i.e., uncorrelated from the circuit errors, they will flip all stabilizers regardless of their value on the pre-measurement state with equal probability. In this case, the measured fidelity satisfies $\hat{F} = F \cdot (1 - 2e_M)$ so that we can deduce the true fidelity $F$ from the measured fidelity and the measurement error. Note that in this case, the measured state fidelity is always a lower bound to the true state fidelity.

**Noisy circuits**

In order to study the influence of experimental noise on the reliability and tightness of our bounds on the TVD, we artificially induce dephasing noise on the 2 × 2 cluster. This simulates a reduced spin-coherence time, which could come from laser phase noise or magnetic field noise. These are the dominant error sources in the experiment. To this end, we pick a fixed instance of the 2 × 2 cluster and add small

random $Z$ rotations on all qubits at roughly equidistant time steps. Specifically, we apply virtual $Z$ gates (i.e., realized in software as an appropriate phase shift on all subsequent gate operations) after the initial local state preparation gates, and again after each MS gate, see Fig. 6. In each run of the experiment (with 50 shots each), we randomly pick rotation angles for the virtual $Z$ gates from a normal distribution with 0 mean and standard deviation $\sigma$. Here $\sigma$ is a measure of the noise strength and corresponds to a local phase-flip probability of $\xi/2$, where $\xi = 1 - e^{-\sigma^2/2}$. If we want to engineer global, correlated noise, we use the same angle for all $Z$ gates in a given "time-step", whereas for engineering local, uncorrelated noise we pick each angle independently. We then average these random choices over 50 instances for the fidelity estimate and 150 instances for the TVD. This averaging turns the random phase shifts into independent (correlated) dephasing channels in the case of local (global) noise. This effectively appears as single-qubit depolarizing noise after every two-qubit gate with a local Pauli error probability of $3\gamma/4$, where $\gamma = 1 - e^{-0.310\sigma^2}$, where the constant was obtained from a numerical fit to simulated data, see Supplementary Note 5.

## Data availability
The data used in this study are available open access at Zenodo: https://doi.org/10.5281/zenodo.13983054.

## Code availability
The code (which uses the Qiskit package[53]) used in the numerical simulations and data analysis can be obtained from the authors.

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

## Acknowledgements

D.H. acknowledges funding from the U.S. Department of Defense through a QuICS Hartree fellowship. This work was completed while D.H. was visiting the Simons Institute for the Theory of Computing. The Berlin team acknowledges funding from the BMBF (DAQC, MUNIQC-ATOMS), the DFG (specifically EI 519/21-1 on paradigmatic quantum devices, but also CRC 183), the Einstein Foundation (Einstein Research Unit), the BMWi (EniQmA, PlanQK), Berlin Quantum, the European Research Council (DebuQC), and the Studienstiftung des Deutschen Volkes. This research is also part of the Munich Quantum Valley (K-8), which is supported by the Bavarian state government with funds from the Hightech Agenda Bayern Plus. It has also received funding from the EU's Horizon 2020 research and innovation program under the Quantum Flagship projects PASQuanS2 and Millenion. J.B.V. acknowledges funding from EU Horizon 2020, Marie Skłodowska-Curie GA. Nr. 754446 - UGR Research and Knowledge Transfer Fund Athenea3i; Digital Horizon Europe project FoQaCiA, GA. Nr. 101070558., and FEDER/Junta de Andalucía program A.FQM.752.UGR20. The Innsbruck team acknowledges support by the Austrian Science Fund (FWF Grant-DOI 10.55776/F71) (SFB BeyondC) and the EU QuantERA project T-NiSQ (I-6001), and the Institut für Quanteninformation GmbH. We also acknowledge funding from the EU H2020-FETFLAG-2018-03 under Grant Agreement No. 820495, by the Office of the Director of National Intelligence (ODNI), Intelligence Advanced Research Projects Activity (IARPA), via US Army Research Office (ARO) grant No. W911NF-21-1-0007, and the US Air Force Office of Scientific Research (AFOSR) via IOE Grant No. FA9550-19-1-7044 LASCEM. This research was funded by the European Union under Horizon Europe Programme—Grant Agreement 101080086—NeQST. Funded by the European Union (ERC, QUDITS, 101039522). Views and opinions expressed are, however, those of the author(s) only and do not necessarily reflect those of the European Union or the European Commission. Neither the European Union nor the granting authority can be held responsible for them.

## Author contributions

M.R. T.F., and L.P. performed the experiments. D.H., M.H., and M.R. analyzed the data and performed the numerical simulations. D.H. and M.H. derived the theory of average (XEB) fidelity in MBQC. D.H., P.K.F., M.H., J.B-V., and J.E. conceived of the protocols. M.R., T.F., C.L.E., C.D.M., R.S., M.M., I.P., L.P., P.S., and T.M. contributed to the experimental setup. R.B., J.E., and T.M. supervised the project. D.H. and M.R. wrote the initial draft of the manuscript. All authors contributed to discussions and writing the final manuscript.

## Competing interests

Thomas Monz, Thomas Feldker, and Rainer Blatt are connected to Alpine Quantum Technologies GmbH, a commercially oriented quantum computing company.
