## [Transparent Peer Review file · Nature Communications]

Verifiable measurement-based quantum random sampling with trapped ions

Corresponding Author: Professor Jens Eisert

Version 0:

Reviewer comments:

Reviewer #1

(Remarks to the Author)

The manuscript by Martin Ringbauer et al reports an experimental demonstration of verifiable quantum random sampling using trapped ions. The main point of the paper is to experimentally implement and test a proposal (Refs. [29,30]) that can effectively verify the sampling problem, which is used to demonstrate the advantages of quantum computers over classical computational performance. Cross-entropy benchmarking (XEB), a previously announced representative verification method for random gate sampling, has a major weakness in that it requires classical simulation of the implemented circuits, which scales exponentially for the computational run-time in estimation. The authors realized a random sampling and verification method based on measurement-based quantum computation, similar to the proposal of Refs. [29,30]. The difference from the previous proposal is that instead of simulating the dynamics of the transverse random Ising model, random cluster states were created directly. After creating a cluster state, one of eight angles was randomly selected and a corresponding z rotation was applied for random sampling. The key to the verification method using cluster states is that the cluster state is completely determined by a small set of stabilizer operators, which allows them to estimate the fidelity of the states with simple measurements. The authors performed a single-instance validation by applying a direct fidelity estimate based on the stabilizer specification and compared it to TVD, then performed an average-case validation by averaging the direct fidelity estimates and compared it to XEB.

This paper reports the first experimental implementation of the theoretically proposed efficient verification proposal of the quantum sampling problem by estimating the fidelity of the cluster state in a simple manner. Although this experiment did not reach quantum advantage, actually far below the level of quantum advantage with cluster states, it did increase the number of qubits in a cluster state to the limits of current experiments. It is also meaningful in that this proposed method was shown to be reliable compared to existing methods. Although the idea was reported a few years ago, I think its publication is meaningful in that its limitations and problems of the scheme can be more clearly revealed through experiments. Therefore, I basically support the publication of this work in Nature Communications. However, I hope that the authors will incorporate the revisions listed below prior to publication. In particular, the paper contains a lot of content, but it is not systematically presented and scattered in many places, making it very difficult to read the paper and understand the main points.

In particular, it is not clearly presented from the beginning of the paper why such efficient verification of random cluster states can reveal quantum advantages based on measurement-based quantum computation (MBQC). It was brought out at the end, the discussion and conclusion section of the paper. MBQC starts from an $n \times m$ cluster state, which can correspond to the m -step operation of n qubits in a circuit model. In other words, for example, performing a 50-depth operation with 100 qubits in a circuit model corresponds to creating a 100×50 cluster state and performing appropriate single qubit rotations and measurements, which is a similar level of quantum computations for circuit model quantum advantages. Because these points were not clearly presented at first, it was difficult to understand how efficient verification of quantum advantage could be achieved by simply generating and measuring a cluster state.

Second, the authors demonstrated experiments on two sets of different trap ion systems. One used recycling methods and the other did not. But it is difficult to find a reason for doing so. First, before looking at the appendix, it was not indicated which data in Figure 3 was obtained through recycling and which data was not. Figures 4 and 5 also do not clearly state which of the two systems was used. Are the two systems used to compare and show that one method is better? There is no direct comparison of the performance of the two experiments, so it is unclear whether such a comparison was intended. It is not clear if this is a claim that efficient verification is possible either way, but the basis is unclear. It is best to describe what

data was obtained and by what method, and clearly state in the text the reasons for using these two methods.

In this regard, I think that related to demonstrating quantum advantages, there should be a discussion not only about the efficient verification but also about whether it is possible to create such a large cluster state with reasonably high fidelity. But I did not find any such discussions at all. There is no discussion about the limits in generating high-fidelity cluster states, or whether such scalability in this direction is possible. How high fidelities do these two methods of generating cluster states provide, and which points can be further improved to reach quantum advantages with cluster states? Why does Figure 5 stop at 2×5 ? Why didn't the authors present how the fidelity changes as the system size increases to their largest one, 4×4 ?

In addition, there are the following questions and suggestions:

1. It may be necessary that the direct fidelity estimation would be described more explicitly. It is not clear from Fig. 1(b) what exactly S_k is. It is not clear what the examples in Figures 1(d,e) are. It appears to be a method based on Ref. [31], but it is better to be self-consistent.
2. Why are both single verification and average verification necessary to show quantum advantages? What different aspects do they reveal for quantum advantages?
3. In the experiment, the cluster state is generated directly, which is different from the previous protocols of Refs. [29,30]. Are they equivalent to showing quantum advantages? Why are the protocols changed in the experimental paper? Some explanation is needed.

Reviewer #2

(Remarks to the Author)

The authors experimentally demonstrate verifiable quantum random sampling in the measurement-based model of quantum computation on a trapped ion quantum processor. In the experiments, they create random cluster states up to a size of 4×4 qubits and verify these states both in single instances and on average based on direct fidelity estimation. Then they compare these results with TVD (total-variation distance) from the classical samples and results based on XEB (cross-entropy benchmarking). It turns out the method based on direct fidelity estimation stands out because it is both sample efficient and computationally efficient. Moreover, it can provide more accurate estimates about the fidelity between the actual state and the ideal state. In the course of study, the authors introduce a method for recycling qubits during the computation, which enables them to sample from entangled cluster states that are larger than the qubit register. This work is of some interest to researchers working on quantum random sampling, quantum advantage, and quantum verification. On the other hand, the significance of the current work is not so compelling at least in the present form, and there are several issues to be addressed.

1. A large proportion of the abstract focuses on the demonstration of quantum advantage based on quantum random sampling and the limitations of existing tools, which leaves the impression that this work is going to demonstrate quantum advantage by overcoming the limitations of previous methods. However, in the end this work does not demonstrate any quantum advantage.
2. Essentially this work created random cluster states and implemented three independent estimation/verification protocols, namely, direct fidelity estimation, TVB estimation, and XEB. All three protocols are well known. Also, it is well known that the fidelity of stabilizer states can be estimated/verified efficiently using local Pauli measurements, which has been demonstrated in a number of experiments. Verification of MBQC in various contexts has also been considered by a number of researchers previously. In view of these facts, the significance of this work beyond previous works is not so clear.
3. It seems the cluster state produced with the recycling method has significantly larger infidelity. Does this problem get more serious as the system size increases? Is this method scalable if high fidelity is desired?
4. The notation "[8]" appeared after Eq.(3) but only explained after Eq.(7).
5. Some journal names in the references are not consistent.

Reviewer #3

(Remarks to the Author)

Dear Editor,

I have read the manuscript "Verifiable measurement-based quantum random sampling with trapped ions" by Martin Ringbauer and co-authors.

The authors consider the setting of randomly rotated cluster states (w.r.t. the z-axis) measured on the X basis. It was shown previously (by some of the present authors) that to predict their outcome distribution accurately is a computationally hard problem for classical computers, yet this is a short-depth process for quantum computers. This paper presents experimental realization on trapped ions of 'efficiently verifiable quantum random sampling' in the measurement-based model. The work is also supported by theory; the Methods and Supplementary Information present further useful theoretical discussions.

They used 40Ca^+ ions as their qubits, and the total number of qubits in the two systems is 4 and 16, respectively. By recycling physical qubits, they extended a physical 2×2 cluster state to a 2×3 cluster state. This experimental technique is

crucial in reaching larger logical systems with limited physical systems.

They used the direct fidelity estimation (DFE) method to measure the single-instance fidelity of a fixed (randomly rotated) cluster state and then the average fidelity of random cluster states. First, they presented the experimental results on the root infidelity (the square root of 1 minus the fidelity) and the total-variation distance (TVD) for single-instance verification. They did this for random cluster states with sizes 1×3 , 2×2 , 2×3 , 3×3 , 4×3 , and 4×4 . The TVD, which is not scalable for large system sizes, is upper bounded by the root infidelity, but there is some substantial gap between the two sets of data.

Then, they studied the effect of noise on the two quantities by introducing local and global phase noise by hand. The gap widens as the strength of the noise increases. The overall behavior is captured by their simulations, and the comparison allowed them to deduce the natural noise floor in the experiment.

Furthermore, they measured two versions of cross-entropy benchmarking (XEB): linear- and log-XEB and used them to estimate the average fidelity. XEB's are not scalable as they require classical simulations to predict ideal probability distributions, which is not feasible as the system size becomes larger. However, the experimental system sizes are still small enough, so the comparison is possible. XEB was compared to the measurement for DFE, and for small sizes 2×2 to 2×4 , the extracted average fidelity was similar, but for 2×5 , the fidelity extracted from XEB differed substantially from DFE.

Their experimental results validate that the MBQC framework provides verifiable random sampling, an alternative to circuit-based random circuit sampling. Moreover, measuring the fidelity for verification is more practical than estimating TVD and XEB. Although the results in this paper do not demonstrate any quantum advantage, the work represents a roadmap towards that. Therefore, I recommend their manuscript be published in Nature Communications.

I do have a few comments for the authors to address to improve their manuscript.

1. In the measurement-based setting presented in this work, what could constitute a demonstration of quantum advantage? What size of cluster states is needed?
2. The authors have a system of 16 qubits, but they did not present results beyond 4×4 . Could the authors comment on that?
3. The authors discussed measurement errors in the Methods. They may also want to consider using readout mitigation.
4. It seems that recycling also introduces noise, as shown in Fig. 3, as the results have larger infidelity. How much can one extend the computational system size compared to the physical size?

Version 1:

Reviewer comments:

Reviewer #1

(Remarks to the Author)

While the revised version does not fully meet my expectations, most of my comments pertain to the presentation of the paper rather than the experimental results or claims. Therefore, I have no major objections to its publication.

I would, however, like to note two points:

The authors stated they have updated the main text and supplementary in response to previous comments, but these changes are not clearly indicated. It would have been helpful if the revisions were highlighted or located explicitly. There appears to be an inconsistency between the MBQC randomized sampling protocol described on page 3 (left column) and the experimental procedure shown in Figure 2. The protocol describes a control-Z operation followed by a single-qubit rotation, while the experiment seems to reverse this order. This discrepancy should be addressed for clarity.

Despite presentation issues, the paper's core content appears sound and suitable for publication.

Reply to Reviewer #1

The manuscript by Martin Ringbauer et al reports an experimental demonstration of verifiable quantum random sampling using trapped ions. The main point of the paper is to experimentally implement and test a proposal (Refs. [29,30]) that can effectively verify the sampling problem, which is used to demonstrate the advantages of quantum computers over classical computational performance. Cross-entropy benchmarking (XEB), a previously announced representative verification method for random gate sampling, has a major weakness in that it requires classical simulation of the implemented circuits, which scales exponentially for the computational run-time in estimation. The authors realized a random sampling and verification method based on measurement-based quantum computation, similar to the proposal of Refs. [29,30]. The difference from the previous proposal is that instead of simulating the dynamics of the transverse random Ising model, random cluster states were created directly. After creating a cluster state, one of eight angles was randomly selected and a corresponding z rotation was applied for random sampling. The key to the verification method using cluster states is that the cluster state is completely determined by a small set of stabilizer operators, which allows them to estimate the fidelity of the states with simple measurements. The authors performed a single-instance validation by applying a direct fidelity estimate based on the stabilizer specification and compared it to TVD, then performed an average-case validation by averaging the direct fidelity estimates and compared it to XEB.

This paper reports the first experimental implementation of the theoretically proposed efficient verification proposal of the quantum sampling problem by estimating the fidelity of the cluster state in a simple manner. Although this experiment did not reach quantum advantage, actually far below the level of quantum advantage with cluster states, it did increase the number of qubits in a cluster state to the limits of current experiments. It is also meaningful in that this proposed method was shown to be reliable compared to existing methods. Although the idea was reported a few years ago, I think its publication is meaningful in that its limitations and problems of the scheme can be more clearly revealed through experiments. Therefore, I basically support the publication of this work in Nature Communications.

We would like to thank the reviewer very much for the thoughtful and detailed report, and for accurately summarizing the main results of our work. We are very happy to read that the reviewer likes our work and “basically support[s]” the publication in Nature Communications.

However, I hope that the authors will incorporate the revisions listed below prior to publication. In particular, the paper contains a lot of content, but it is not systematically presented and scattered in many places, making it very difficult to read the paper and understand the main points.

We are very happy, once more, to get such detailed feedback on what we can do to improve the presentation of our work. We have taken these comments very seriously. In the following, we explain what we have done to accommodate them.

In particular, it is not clearly presented from the beginning of the paper why such efficient verification of random cluster states can reveal quantum advantages based on measurement-based quantum computation (MBQC). It was brought out at the end, the discussion and conclusion section of the paper. MBQC starts from an $n \times m$ cluster state, which can correspond to the m -step operation of n qubits in a circuit model. In other words, for example, performing a 50-depth operation with 100 qubits in a circuit model corresponds to creating a 100×50 cluster state and performing appropriate single qubit rotations and measurements, which is a similar level of quantum computations for circuit model quantum advantages. Because these points were not clearly presented at first, it was difficult to understand how efficient verification of quantum advantage could be achieved by simply generating and measuring a cluster state.

We thank the reviewer very much for this comment. Rereading the manuscript from some distance, we agree with you that this has not been completely clear. We have now rewritten the section on “Sampling

and verification protocols” and hope that the relationship between a circuit-model-based approach to random sampling and the measurement-based approach comes across clearly now.

Second, the authors demonstrated experiments on two sets of different trap ion systems. One used recycling methods and the other did not. But it is difficult to find a reason for doing so. First, before looking at the appendix, it was not indicated which data in Figure 3 was obtained through recycling and which data was not. Figures 4 and 5 also do not clearly state which of the two systems was used. Are the two systems used to compare and show that one method is better? There is no direct comparison of the performance of the two experiments, so it is unclear whether such a comparison was intended. It is not clear if this is a claim that efficient verification is possible either way, but the basis is unclear. It is best to describe what data was obtained and by what method, and clearly state in the text the reasons for using these two methods.

We appreciate that there might be some confusion about the use of two devices. Indeed, we consider the fact that we used two different physical devices rather unimportant and certainly did not intend for this paper to present a comprehensive comparison between the two devices. Instead, we simply intended to supplement the standard approach, which we used to generate a 16-qubit cluster state with a proof-of-principle of an advanced approach of generating cluster states on the fly. The latter approach requires different experimental capabilities in terms of optical access and field orientations to enable mid-circuit recooling. Moreover, the recooling does not reach the motional ground state, as we indicated in our original manuscript. Hence, the recycling experiment comes with somewhat increased noise levels at the current stage of development, yet has immense potential for reducing the number of qubits required for measurement-based quantum random sampling. In our revision, we have highlighted the proof-of-principle character of the recycling demonstration and clarified that a comparison is not intended.

In this regard, I think that related to demonstrating quantum advantages, there should be a discussion not only about the efficient verification but also about whether it is possible to create such a large cluster state with reasonably high fidelity. But I did not find any such discussions at all. There is no discussion about the limits in generating high-fidelity cluster states, or whether such scalability in this direction is possible. How high fidelities do these two methods of generating cluster states provide, and which points can be further improved to reach quantum advantages with cluster states? Why does Figure 5 stop at 2×5 ? Why didn't the authors present how the fidelity changes as the system size increases to their largest one, 4×4 ?

We thank the reviewer for raising this very relevant point. Indeed, the cluster-state fidelity needs to be sufficiently high for the sampling task to be classically hard. The fidelity threshold which is provable is indicated by the gray shaded area in Fig. 3, but generally, we expect the required fidelity for hardness to just be bounded away from 2^{-n} as we discuss in the manuscript, which we emphasize in the revised main text. Hence, the sampling protocol requires that the gate error decreases with cluster state size in order to remain classically hard. We note, however, that this is a requirement for practically any useful quantum computation. The crucial aspect is that the resource requirements for the verification of these states do not scale with the qubit number.

Since the requirements for on-the-fly cluster state generation are a subset of those required for quantum error correction, it is fair to assume that, in the long run, the two methods must perform roughly equally, albeit with different qubit number requirements. Recent QEC demonstrations (e.g. Phys. Rev. X 11, 041058) further support the expectation that the technical limitations of our proof-of-principle qubit recycling approach can be overcome and similar performance can be achieved.

Finally, the results of Fig. 5 were meant to investigate the scaling with the “depth” of the computation, since the quality of XEB as a fidelity estimator in the circuit model depends on the circuit depth, and—there—is expected to work better for deeper circuits (at least $\sim \log n$). Hence, it is relevant to investigate the quality of fidelity estimation depending on the relative dimensions of the cluster states (corresponding to the depth of the circuit). A finding of the figure is that there are some deviations but no significant

dependence on the dimension. The measured points are sufficient to illustrate this finding. We acknowledge that this did not become clear previously, but we have clarified it now in our discussion of the results.

In addition, there are the following questions and suggestions:

1. It may be necessary that the direct fidelity estimation would be described more explicitly. It is not clear from Fig. 1(b) what exactly S_k is. It is not clear what the examples in Figures 1(d,e) are. It appears to be a method based on Ref. [31], but it is better to be self-consistent.

Thank you very much for pointing this out. We have amended the caption to clarify the definition of S_k in Fig 1(d,e). We further extended our explanation of the DFE protocol in the main text, referring to the appropriate Methods section for the full technical details, which would require too much space in the main text.

In brief, in direct fidelity estimation, we draw a random element of the (exponentially large) stabilizer group of the target state, measure the corresponding operator, and then average the ± 1 outcomes to obtain the fidelity estimate. By linearity, this allows us to also estimate the fidelity averaged over an ensemble with an error that concentrates quickly and independently of the system size.

2. Why are both single verification and average verification necessary to show quantum advantages? What different aspects do they reveal for quantum advantages?

Thank you for pointing this out. We have now clarified in the introduction that single-instance fidelity estimation allows us to certify samples from a fixed cluster state, which certifies a fixed distribution, whereas average-fidelity estimation yields an average performance benchmark of the device preparing random states. Thus, the two serve a somewhat different purpose; we think of the former as verification of a distribution, and of the latter as benchmarking the device.

3. In the experiment, the cluster state is generated directly, which is different from the previous protocols of Refs. [29,30]. Are they equivalent to showing quantum advantages? Why are the protocols changed in the experimental paper? Some explanation is needed.

We have now clarified after introducing the protocol on page 3 that our gate-based way to prepare the cluster state is equivalent to the formulation of Refs. [29,30] ([26,27] in the new count), where the focus lies on analog time evolution. The reason for the different formulations is that it teaches us different things: when we think of the state preparation via analog time evolution, our protocol shows that analog time evolution shows quantum advantage. When we think of it in the measurement-based model, it lets us make statements about random computations in that model, which is the focus of our work.

Reply to Reviewer #2

The authors experimentally demonstrate verifiable quantum random sampling in the measurement-based model of quantum computation on a trapped ion quantum processor. In the experiments, they create random cluster states up to a size of 4×4 qubits and verify these states both in single instances and on average based on direct fidelity estimation. Then they compare these results with TVD (total-variation distance) from the classical samples and results based on XEB (cross-entropy benchmarking). It turns out the method based on direct fidelity estimation stands out because it is both sample efficient and computationally efficient. Moreover, it can provide more accurate estimates about the fidelity between the actual state and the ideal state. In the course of study, the authors introduce a method for recycling qubits during the computation, which enables them to sample from entangled cluster states that are larger than the qubit register. This work is of some interest to researchers working on quantum random sampling, quantum advantage, and quantum verification.

We would like to thank the reviewer for the helpful and thoughtful report.

On the other hand, the significance of the current work is not so compelling at least in the present form, and there are several issues to be addressed.

1. A large proportion of the abstract focuses on the demonstration of quantum advantage based on quantum random sampling and the limitations of existing tools, which leaves the impression that this work is going to demonstrate quantum advantage by overcoming the limitations of previous methods. However, in the end this work does not demonstrate any quantum advantage.

We thank the reviewer for this comment. The focus of our manuscript is to demonstrate a scalable method for the verification of quantum random sampling experiments. The motivation for developing such methods is to use them in demonstrations of a quantum advantage as our abstract explicitly states. In the abstract, we never claim to demonstrate quantum advantage. It explicitly mentions the largest used cluster state size, which is clearly below the quantum advantage regime and concludes with describing our work as “a feasible path toward a verified demonstration of a quantum advantage.” Hence, we believe our abstract is accurate in that it discusses quantum advantage experiments and their verification as a leading motivation for our work, but makes clear that our experiments are a proof of principle.

2. Essentially this work created random cluster states and implemented three independent estimation/verification protocols, namely, direct fidelity estimation, TVB estimation, and XEB. All three protocols are well known. Also, it is well known that the fidelity of stabilizer states can be estimated/verified efficiently using local Pauli measurements, which has been demonstrated in a number of experiments. Verification of MBQC in various contexts has also been considered by a number of researchers previously. In view of these facts, the significance of this work beyond previous works is not so clear.

We thank the reviewer for this comment. Indeed, direct fidelity estimation (DFE) has been known for a while, but it was to the best of our knowledge not clear that it could be used to verify a quantum advantage. Indeed, Leone *et al.* [Phys. Rev. A 107, 022429 (2023)] recently argued that standard DFE requires a number of samples that scales exponentially in the amount of non-Cliffordness in the state. However, a large amount of non-Cliffordness is also required for quantum advantage, which would suggest that DFE cannot be used to verify quantum advantage. Our observation is that in fact a variant of DFE can be used to efficiently verify the fidelity of large-scale cluster states with a large amount of non-Cliffordness, and that these states are in fact sufficient for quantum advantage demonstrations. We also observe that it can naturally give average fidelity estimates. We believe that both of these observations are novel and noteworthy.

All other protocols, specifically XEB and TVD estimation, which are non-scalable due to the use of classical samples, are compared to the DFE estimates in order to demonstrate that — for the purpose of fidelity

estimation — DFE is in all measures superior to the current standard means of verifying quantum advantage: DFE gives more accurate estimates of the fidelity and is completely efficient, whereas XEB is often not as accurate and requires exponential computing time.

Moreover, we also demonstrate that the previously proposed method for efficient verification of measurement-based random sampling – fidelity witnessing (Bermejo-Vega *et al.*, Phys. Rev. X (2018)) – will not be feasible at large scales, since even at the small scales investigated here, the resulting certificate is much too loose to be useful (see, SM Fig S1). Thus, the new DFE protocol is a necessary novel ingredient for verified quantum advantage in the measurement-based setting.

Finally, from the experimental perspective, we for the first time demonstrate in a proof-of-principle experiment how (random) cluster states can be generated on the fly using qubit recycling. This highlights that one of the main drawbacks of MBQC, namely the high qubit count, can be overcome in future experiments.

We, therefore, believe that our work is the first to provide a clear and feasible path towards verified quantum advantage. It does so by developing a new approach to verifying random cluster states based on modified direct fidelity estimation, introducing and demonstrating the use of qubit recycling in order to generate large clusters, and overall demonstrating the experimental feasibility of the proposed methods. We have clarified these contributions in the introduction of the revised version of the manuscript.

3. It seems the cluster state produced with the recycling method has significantly larger infidelity. Does this problem get more serious as the system size increases? Is this method scalable if high fidelity is desired?

The recycling method was intended as a proof-of-principle demonstration of a much more efficient approach to generating random cluster states for sampling. With the required experimental techniques still in the early stages, the resulting fidelities are somewhat lower due to technical limitations. However, despite the increased technical difficulties, there are no fundamental reasons for the recycling method to perform any worse in terms of fidelities than the standard approach. To the contrary, we expect the quadratically reduced qubit count to positively affect the fidelities in the long run. We clarified these points in our revision.

4. The notation “[8]” appeared after Eq.(3) but only explained after Eq.(7).

We thank the reviewer for spotting this. We have moved the definition to the first time the notation appears.

5. Some journal names in the references are not consistent.

We thank the reviewer very much for the comment. We have cleaned up the references. Again, we thank the reviewer for the careful report and hope that our work is now suitable for publication.

Reply to Reviewer #3

Dear Editor,

I have read the manuscript "Verifiable measurement-based quantum random sampling with trapped ions" by Martin Ringbauer and co-authors.

The authors consider the setting of randomly rotated cluster states (w.r.t. the z-axis) measured on the X basis. It was shown previously (by some of the present authors) that to predict their outcome distribution accurately is a computationally hard problem for classical computers, yet this is a short-depth process for quantum computers. This paper presents experimental realization on trapped ions of 'efficiently verifiable quantum random sampling' in the measurement-based model. The work is also supported by theory; the Methods and Supplementary Information present further useful theoretical discussions.

They used 40Ca^+ ions as their qubits, and the total number of qubits in the two systems is 4 and 16, respectively. By recycling physical qubits, they extended a physical 2×2 cluster state to a 2×3 cluster state. This experimental technique is crucial in reaching larger logical systems with limited physical systems.

They used the direct fidelity estimation (DFE) method to measure the single-instance fidelity of a fixed (randomly rotated) cluster state and then the average fidelity of random cluster states. First, they presented the experimental results on the root infidelity (the square root of 1 minus the fidelity) and the total-variation distance (TVD) for single-instance verification. They did this for random cluster states with sizes 1×3 , 2×2 , 2×3 , 3×3 , 4×3 , and 4×4 . The TVD, which is not scalable for large system sizes, is upper bounded by the root infidelity, but there is some substantial gap between the two sets of data.

Then, they studied the effect of noise on the two quantities by introducing local and global phase noise by hand. The gap widens as the strength of the noise increases. The overall behavior is captured by their simulations, and the comparison allowed them to deduce the natural noise floor in the experiment.

Furthermore, they measured two versions of cross-entropy benchmarking (XEB): linear- and log-XEB and used them to estimate the average fidelity. XEB's are not scalable as they require classical simulations to predict ideal probability distributions, which is not feasible as the system size becomes larger. However, the experimental system sizes are still small enough, so the comparison is possible. XEB was compared to the measurement for DFE, and for small sizes 2×2 to 2×4 , the extracted average fidelity was similar, but for 2×5 , the fidelity extracted from XEB differed substantially from DFE.

Their experimental results validate that the MBQC framework provides verifiable random sampling, an alternative to circuit-based random circuit sampling. Moreover, measuring the fidelity for verification is more practical than estimating TVD and XEB. Although the results in this paper do not demonstrate any quantum advantage, the work represents a roadmap towards that. Therefore, I recommend their manuscript be published in Nature Communications.

We thank the reviewer for their detailed reading and constructive feedback.

I do have a few comments for the authors to address to improve their manuscript.

1. In the measurement-based setting presented in this work, what could constitute a demonstration of quantum advantage? What size of cluster states is needed?

In the revised manuscript, we are much more careful in relating the measurement-based and the circuit model of quantum computation, which we believe will clarify the relationship. We are hesitant to give concrete numbers in the manuscript since we have not analyzed actual numerical simulation algorithms, we

expect that cluster states of 50x50 to 100x100 qubits should be sufficient for quantum advantage, since the effective circuit implemented by such cluster states acts on 50 to 100 qubits.

2. The authors have a system of 16 qubits, but they did not present results beyond 4x4. Could the authors comment on that?

The large cluster results were obtained on a device with 16 qubits, whereas the recycling method was a proof-of-principle demonstration on a different, smaller device that satisfied the technical requirements for qubit recycling. In our revised manuscript, we have changed the wording accordingly to make this clear.

3. The authors discussed measurement errors in the Methods. They may also want to consider using readout mitigation.

We thank the reviewer for this suggestion. Our analysis in the methods shows that (non-adversarial) readout errors lead to a decrease in the measured fidelity compared to the true value and do therefore not compromise the faithfulness of the protocol. To the best of our knowledge, readout mitigation works in expectation, but not on a shot-by-shot basis as required in the randomized measurement scheme employed in our work. Hence, more research is needed before a rigorous statement about the use of readout mitigation can be made.

4. It seems that recycling also introduces noise, as shown in Fig. 3, as the results have larger infidelity. How much can one extend the computational system size compared to the physical size?

The reviewer is correct in the observation that the recycling method comes with some added noise due to technical limitations in this proof-of-principle demonstration of on-the-fly cluster state generation. There are no fundamental limits beyond the standard approach and as such we expect them to achieve comparable fidelities in the long run, albeit using different qubit numbers. The degree to which the computational size can be extended compared to the physical size will then depend on the infidelity that can be tolerated.

Reply to Referee 1

Thank you very much for the final report. We reply here to the one comment which concerns the content of the manuscript.

There appears to be an inconsistency between the MBQC randomized sampling protocol described on page 3 (left column) and the experimental procedure shown in Figure 2. The protocol describes a control-Z operation followed by a single-qubit rotation, while the experiment seems to reverse this order. This discrepancy should be addressed for clarity.

Thank you for this comment. Indeed, the ideal protocol needs to be compiled to the natural gate set of the ion-trap processor, as we detail in Supplementary Note 3. This section is referenced both before the protocol description on p. 3 and in the caption of Fig. 2. In particular, since the single-qubit rotations commute with the entangling gates, it does not matter whether they are applied before or after them.